# Diffusion Visual Counterfactual Explanations

**Maximilian Augustin**[*]   **Valentyn Boreiko**[*]   **Francesco Croce**   **Matthias Hein**
University of Tübingen

## Abstract

Visual Counterfactual Explanations (VCEs) are an important tool to understand the decisions of an image classifier. They are "small" but "realistic" semantic changes of the image changing the classifier decision. Current approaches for the generation of VCEs are restricted to adversarially robust models and often contain non-realistic artefacts, or are limited to image classification problems with few classes. In this paper, we overcome this by generating Diffusion Visual Counterfactual Explanations (DVCEs) for arbitrary ImageNet classifiers via a diffusion process. Two modifications to the diffusion process are key for our DVCEs: first, an adaptive parameterization, whose hyperparameters generalize across images and models, together with distance regularization and late start of the diffusion process, allow us to generate images with minimal semantic changes to the original ones but different classification. Second, our cone regularization via an adversarially robust model ensures that the diffusion process does not converge to trivial non-semantic changes, but instead produces realistic images of the target class which achieve high confidence by the classifier. Code is available under `https://github.com/valentyn1boreiko/DVCEs`.

## 1   Introduction

It can be argued that one of the main problems hindering the widespread use of machine learning, and image classification in particular, is the missing possibility to explain the decisions of black-box models such as neural networks. This is not only a problem for decisions affecting humans where the current draft for AI regulation in Europe [10] requires "transparency", but it is a pressing problem in all applications of machine learning. The reason, to some extent, is that humans would like to understand but also control if the learning algorithm has captured the "concepts" of the underlying classes or if it just predicts well using spurious features, artefacts in the data set, or other sources of error. In this paper, we focus on model-agnostic explanations which can, in principle, be applied to any image classifier and do not rely on the specific structure of the classifier such as decision trees or linear classifiers. In this area, in particular for image classification, sensitivity based explanations [4], explanations based on feature attributions [3], saliency maps [47, 46, 17, 56, 51], Shapley additive explanations [31], and local fits of interpretable models [39] have been proposed. Moreover, [55] proposed counterfactual explanations (CEs), which are instance-specific explanations. They can be applied to any classifier and it has been argued that they are close to the human justification of decisions [34] using counterfactual reasoning: "I would recognize it as zebra (instead of horse) if it had black and white stripes." For a given classifier almost all methods for the generation of CEs [55, 15, 35, 6, 38, 54, 44] try to solve the following problem: "Given a target class $c$ what is the minimal change $\delta$ of input $x$, such that $x + \delta$ is classified as class $c$ with high probability and is a realistic instance of my data generating distribution?" From the perspective of "debugging" existing machine learning models, CEs are interesting as they construct a concrete input $x + \delta$ with a different classification that allows the developer to test if the model has learned the correct features.

---

[*]Equal contribution.

| Original | Class 1 | Class 2 | Original | Class 1 | Class 2 |
|----------|---------|---------|----------|---------|---------|
| chimpanzee 0.88 | orangutan: 0.97 | gorilla: 0.97 | Chesapeake Bay ret.: 0.97 | golden retriever: 0.93 | labrador retriever: 0.67 |

Figure 1: Our method DVCE creates visual counterfactual explanations (VCEs) that explain any given classifier (here: non-robust ConvNeXt) by changing an image of one class into a similar target class by changing class relevant features while preserving the overall image structure.

The main reason why visual counterfactual explanations (VCEs), that is CEs for image classification, are not widely used is that the tasks of generating CEs and adversarial examples [52] are very related. Even imperceivable changes of the image can already change the prediction of the classifier, however, the resulting noise patterns do not show the user if the classifier has picked up the right class-specific features. One possible solution is using adversarially robust models [43, 1, 7], which have been shown to produce semantically meaningful VCEs by directly maximizing the probability of the target class in image space. These approaches have the downside that they only can generate VCEs for robust models which is a significant restriction as these models are not competitive in terms of prediction accuracy. The second approach is to restrict the generation of VCEs using a generative model or constraining the set of potential image manipulations [19, 20, 42, 8, 18, 45, 27]. However, these approaches are either restricted to datasets with a small number of classes, cannot provide explanations for arbitrary classifiers, or generate VCEs that look realistic but have so little in common with the original image that not much insight can be gained.

Recently, [26] trained a StyleGAN2 [24] model to discover and manipulate class attributes. While their approach yields impressive results on smaller-scale datasets with few similar classes (for example, different bird species), the authors did not demonstrate that the method scales to complex tasks such as ImageNet with hundreds of classes where different classes require different sets of attributes. Another disadvantage is that the StyleGAN model needs to be retrained for every classifier, making it prohibitively expensive to explain multiple large models. Moreover, [25] have proposed a loss to do VCEs in the latent space of a GAN/VAE. They show promising results for MNIST/FMNIST, but no code is available for ImageNet. Furthermore, to explain a classifier, the conditional information during the generation of explanations should ideally come only from the classifier itself, but [25] rely on a conditional GAN which might introduce a bias.

In this paper, we overcome the aforementioned challenges and generate our Diffusion Visual Counter-factual Explanations (DVCEs) for arbitrary ImageNet classifiers (see Fig 1). We use the progress in the generation of realistic images using diffusion processes [48, 23, 49, 50], which recently were able to outperform GANs [14]. Similar to [37, 2], we use a classifier and a distance-type regularization to guide the generation of the images. Two modifications to the diffusion process are key elements for our DVCEs: i) a combination of distance regularization and starting point of the diffusion process together with an adaptive reparameterization lets us generate VCEs visually close to the original image in a controlled way so that hyperparameters can be fixed across images and even across models, ii) our cone regularization of the gradient of the classifier via an adversarially robust model ensures that the diffusion process does not converge to trivial non-semantic changes but instead produces realistic images of the target class which achieve high confidence by the classifier. Our approach can be employed for any dataset where a generative diffusion model and an adversarially robust model are available. In a qualitative comparison, user study, and a quantitative evaluation (Sec. 4.1), we show that our DVCEs achieve higher realism (according to FID) and have more meaningful features (according to the user study) than both the recent methods of [7] and [2].

## 2 Diffusion models

Diffusion models [14, 48, 49, 23, 36] are generative models that consist of two steps: a *forward diffusion* process (and thus a Markov process) that transforms the data distribution to a prior distribution (which is usually assumed to be a standard normal distribution) and the *reverse diffusion* process that

transforms the prior distribution back to the data distribution. The existence of the (continuous-time) reverse diffusion process was closer investigated in [50].

In the discrete-time setting, a Markov chain $\{x_1, ..., x_T\}$ for any data point $x_0$ in the forward direction is defined via a Markov process that adds noise to the data point at each timestep:

$$q(x_t|x_{t-1}) = \mathcal{N}(\sqrt{1-\beta_t}x_{t-1}, \beta_t I), \tag{1}$$

where $\{\beta_1, ..., \beta_T\}$ is some variance schedule, chosen such that $q(x_T) \sim \mathcal{N}(0, I)$. Note that given $x_0$, it is possible to sample from $q(x_t|x_0)$ in closed form instead of applying noise $t$ times using:

$$q(x_t|x_0) = \mathcal{N}(\sqrt{\overline{\alpha}_t}x_0, 1 - \overline{\alpha}_t I), \tag{2}$$

$$x_t = \sqrt{\overline{\alpha}_t}x_0 + \epsilon\sqrt{1-\overline{\alpha}_t}, \quad \epsilon \sim \mathcal{N}(0, I), \tag{3}$$

where $\overline{\alpha}_t := \prod_{k=1}^{t}(1-\beta_k)$. In [48], the authors have shown that the reverse transitions $q(x_{t-1}|x_t)$ approach diagonal Gaussian distributions as $T \to \infty$ and thus one can use a DNN with parameters $\theta$ to approximate $q(x_{t-1}|x_t)$ as $p_\theta(x_{t-1}|x_t)$ by predicting the mean and the diagonal covariance matrix $\mu_\theta(x_t, t)$ and $\Sigma_\theta(x_t, t)$:

$$p_\theta(x_{t-1}|x_t) = \mathcal{N}(\mu_\theta(x_t, t), \Sigma_\theta(x_t, t)). \tag{4}$$

To sample from $p_\theta(x_0)$, one samples $x_T$ from $\mathcal{N}(0, I)$ and then follows the reverse process by repeatedly sampling from the transition probabilities $p_\theta(x_{t-1}|x_t)$.

Instead of predicting $\mu_\theta(x_t, t)$ directly, it has been shown in [23] that the best performance can be achieved when a neural network $\epsilon_\theta(x_t, t)$ predicts the source noise $\epsilon$ in (3). Then, the loss used for training resembles that of a denoising model:

$$L_{\text{simple}}(\theta, T, q(x_0)) := \mathbb{E}_{t\sim[1,T], x_0\sim q(x_0), \epsilon\sim\mathcal{N}(0,I)} \|\epsilon - \epsilon_\theta(x_t, t)\|^2. \tag{5}$$

The mean of the reverse step $\mu_\theta$ in (4) can be derived [32] using Bayes theorem as:

$$\mu_\theta(x_t, t) = \frac{1}{\sqrt{1-\beta_t}}\Big(x_t - \frac{\beta_t}{\sqrt{1-\overline{\alpha}_t}}\epsilon_\theta(x_t, t)\Big). \tag{6}$$

This is, however, only one of the possible equivalent parameterization of the learning objective as shown in [32]. Because the objective $L_{\text{simple}}$ does not give a learning signal for $\Sigma_\theta(x_t, t)$, in practice one combines $L_{\text{simple}}$ with another loss based on a variational lower bound of the data likelihood [48], which unlike $L_{\text{simple}}$ allows us to learn the diagonal covariance matrix $\Sigma_\theta(x_t, t)$. Concretely, the network $\epsilon_\theta(x_t, t)$ outputs additionally a vector that is used to parametrize $\Sigma_\theta(x_t, t)$.

## 2.1 Class conditional sampling

There exist diffusion models trained with and without knowledge about classes in the dataset [14, 22]. For our experiments, we only use the class-unconditional diffusion model, such that all class-conditional features are introduced by the classifier that we want to explain. For the noise-aware classifier $p_\phi(y|\cdot, \cdot) : \mathbb{R}^d \times \{1, ..., T\} \to [0, 1]$, with parameters $\phi$, that is trained on noisy images corresponding to the various timesteps [14], the reverse process transitions are then of the form:

$$p_{\theta,\phi}(x_{t-1}|x_t, y) = Z\, p_\theta(x_{t-1}|x_t)\, p_\phi(y|x_t, t) \tag{7}$$

for a normalization constant $Z$. As we would like to explain any classifier and not only noise-aware ones, we follow the approach of [2], where a classifier $p_\phi(y|\cdot) : \mathbb{R}^d \to [0, 1]$ is given as input the denoised sample $\hat{x}_0 = f_{\text{dn}}(x_t)$ of $x_t$, using the mapping:

$$f_{\text{dn}} : \mathbb{R}^d \times \{1, ..., T\} \to \mathbb{R}^d, \quad (x_t, t) \mapsto \frac{x_t}{\sqrt{\overline{\alpha}_t}} - \frac{\sqrt{1-\overline{\alpha}_t}\epsilon_\theta(x_t, t)}{\sqrt{\overline{\alpha}_t}}. \tag{8}$$

This mapping, derived from (3), estimates the noise-free image $x_0$ using the noise approximated by the model $\epsilon_\theta(x_t, t)$ for a given timestep $t$. With this, we can define a timestep-aware posterior $p_\phi(y|x_t, t)$ for any classifier $p_\phi(y|\cdot)$ as $p_\phi(y|x_t, t) := p_\phi(y|f_{\text{dn}}(x_t, t))$.

The issue is that efficient sampling from the original diffusion model is possible only because the reverse process is made of normal distributions. As we need to sample from $p_{\theta,\phi}(x_{t-1}|x_t, y)$ hundreds

of times to obtain a single sample from the data distribution, it is not possible to use MCMC-samplers with high complexity to sample from each of the individual transitions. In [14], they proposed to solve it by approximating $p_{\theta,\phi}(x_{t-1}|x_t, y)$ with slightly shifted versions of $p_\theta(x_{t-1}|x_t)$ to make closed-form sampling possible. Such transition kernels are given by:

$$p_{\theta,\phi}(x_{t-1}|x_t, y) = \mathcal{N}(\mu_t, \Sigma_\theta(x_t, t)), \tag{9}$$

$$\mu_t = \mu_\theta(x_t, t) + \Sigma_\theta(x_t, t)\nabla_{x_t} \log p_\phi(y|x_t, t), \tag{10}$$

which we further adapt for the goal of generating VCEs and use in our experiments.

## 3 Diffusion Visual Counterfactual Explanations

A VCE $x$ for a chosen target class $y$, a given classifier $p_\phi(y|\cdot)$, and an input $\hat{x}$ should satisfy the following criteria: i) **validity:** the VCE $x$ should be classified by $p_\phi(y|\cdot)$ as the desired target class $y$ with high predicted probability, ii) **realism:** the VCE should be as close as possible to a natural image, iii) **minimality/closeness:** the difference between the VCE $x$ and the original image $\hat{x}$ should be the minimal semantic modification necessary to change the class, in particular, the generated image $x$ should be close to $\hat{x}$ while being valid and realistic, e.g. by changing the object in the image and leaving the background unchanged. Note that targeted adversarial examples are valid but do not show meaningful semantic changes in the target class for a non-robust model and are not realistic. The $l_{1.5}$-SVCEs of [7] change the image in order to maximize the predicted probability of the classifier into the target class inside an $l_{1.5}$-ball around the image which is a targeted adversarial example. Thus this only works for robust classifiers and they use an ImageNet classifier that was trained to be multiple-norm robust (MNR), which we denote in this paper as MNR-RN50 (see Sec. 4.2). The realism of the $l_{1.5}$-SVCEs comes purely from the generative properties of robust classifiers [43], which can lead to artefacts. In contrast, our Diffusion Visual Counterfactual Explanations (DVCEs) work for any classifier and our DVCEs are more realistic due the better generative properties of diffusion models. An approach similar to our DVCE framework is Blended Diffusion (BD) [2] which manipulates the image inside a masked region. One can adapt BD for the generation of VCEs by using as mask the whole image. DVCE and BD share the same diffusion model, but BD cannot be applied to arbitrary classifiers and requires image-specific hyperparameter tuning, see Fig. 2.

### 3.1 Adaptive Parameterization

For generating the DVCE of the original image $\hat{x}$, we do not want to sample just any image $x$ from $p(x|y)$ (high realism and validity), but also to make sure that $d(x, \hat{x})$ is small. For this, we have to condition our diffusion process on $\hat{x}$. Using the denoising function $f_{\mathrm{dn}}$ and analogously to the derivation of (9) and its applications in [14, 27], the mean of the transition kernel becomes:

$$\mu_\theta(x_t, t) + \Sigma_\theta(x_t, t)\nabla_{x_t}\Big[C_c \log p_\phi\big(y|f_{\mathrm{dn}}(x_t, t)\big) - C_d\, d\big(\hat{x}, f_{\mathrm{dn}}(x_t, t)\big)\Big], \tag{11}$$

where $C_c$ is the coefficient of the classifier, and $C_d$ the one of the distance guiding loss. Intuitively, we take a step in the direction that increases the classifier score while staying close to $\hat{x}$. Note that the last term can be interpreted as the log gradient of a distribution with density $\exp(-C_d\, d_t(\hat{x}, \cdot))$, where $d_t(\hat{x}, \cdot) := d(\hat{x}, f_{\mathrm{dn}}(\cdot, t))$. Thus we introduce a timestep-aware prior distribution that enforces our output to be similar to $\hat{x}$. In our work, we use the $l_1$-distance as it produces sparse changes.

Several works have tried to minimize some distance during the diffusion process, implicitly in [9] or explicitly, but for the background, in BD. However, there is no principled way to choose the coefficient for such a regularization term. It turns out that it is impossible to find a parameter setting for BD that works across images and classifiers. Thus, we propose a parameterization

$$g_{\mathrm{update}} = C_c \frac{\nabla_{x_t} \log p_\phi(y|f_{\mathrm{dn}}(x_t, t)))}{\|\nabla_{x_t} \log p_\phi(y|f_{\mathrm{dn}}(x_t, t)))\|_2} - C_d \frac{\nabla_{x_t} d(\hat{x}, f_{\mathrm{dn}}(x_t, t))}{\|\nabla_{x_t} d(\hat{x}, f_{\mathrm{dn}}(x_t, t))\|_2}, \tag{12}$$

which adapts to the predicted mean of the diffusion model that we use to change $\mu_t$ in (10) to

$$\mu_t = \mu_\theta(x_t, t) + \Sigma_\theta(x_t, t)\, \|\mu_\theta(x_t, t)\|_2\, g_{\mathrm{update}}. \tag{13}$$

This adaptive parameterization allows for fine-grained control of the influence of the classifier and distance regularization so that now the hyperparameters $C_c$ and $C_d$ have the same influence across

| Original | DVCEs | BDVCEs $C_c$ 10 | | | BDVCEs $C_c$ 25 | | |
|---|---|---|---|---|---|---|---|
| | | $C_d = 100$ | 500 | 1000 | 100 | 500 | 1000 |
| keeshond: 0.38 | chow: 1.00 | chow: 0.77 | chow: 0.55 | chow: 0.07 | chow: 0.99 | chow: 0.64 | chow: 0.80 |
| lynx: 0.72 | cheetah: 0.99 | cheetah: 0.67 | cheetah: 0.38 | cheetah: 0.77 | cheetah: 0.95 | cheetah: 0.73 | cheetah: 0.98 |
| weasel: 0.41 | guinea pig: 1.00 | guinea pig: 0.86 | guinea pig: 0.12 | guinea pig: 0.05 | guinea pig: 0.93 | guinea pig: 0.97 | guinea pig: 0.70 |

Figure 2: DVCEs for the robust MNR-RN50 model [7] in the second column (ours) and using Blended Diffusion (BDVCEs) [2], such that regularization is applied to the whole image. Due to our adaptive parameterization, the same parameters ($C_c = 0.1, C_d = 0.15$) work for our DVCEs across different images and classifiers (see Fig. 5). For BDVCEs, it is difficult to choose a set of hyperparameters that, even for a single classifier, work for multiple images. For BDVCEs, we report the parameter $C_d$ of the LPIPS weight and the $l_2$-weight is 10 times as large (ratio used in [2]).

images and even classifiers. It facilitates the generation of DVCEs as otherwise hyperparameter finetuning would be necessary for each image as in BD, see Fig. 2 for a comparison. However, even with our adaptive parameterization, it is still not easy to produce semantically meaningful changes close to $\hat{x}$ as can be seen in App. B.2. Thus, as in [2], we vary the starting point of the diffusion process and observe in App. B.2 that starting from step $\frac{T}{2}$ of the forward diffusion process, together with the adaptive parameterization and using as the distance the $l_1$-distance, provides us with sparse but semantically meaningful changes. In our experiments, we set $T = 200$.

## 3.2 Cone Projection for Classifier Guidance

A key objective for our DVCEs is that they can be applied to any image classifier, adversarially robust or not. Diffusion with the new mean as in (13) does not work with a non-robust classifier and leads to very small modifications of the image, which are similar to adversarial examples without meaningful changes. The reason is that the gradients of non-robust classifiers are noisier and less semantically meaningful than those of robust classifiers. We illustrate this for a non-robust Swin-TF [28] in Fig. 3 where hardly any changes are generated. This even holds if we first denoise the sample using $f_{\text{dn}}$ from (8) and average gradients of augmented images as in [2] (see Appendix B.6 for details).

As a solution, we suggest projecting the gradient of an adversarially robust classifier with parameters $\psi$, $\nabla_{x_t} \log p_{\text{robust},\psi}(y|f_{\text{dn}}(x_t,t))$, onto a cone centered at the gradient of the classifier, $\nabla_{x_t} \log p_\phi(y|f_{\text{dn}}(x_t,t))$. More precisely, we define

$$g_{\text{proj}} = P_{\text{cone}(\alpha,\nabla_{x_t} \log p_\phi(y|f_{\text{dn}}(x_t,t)))}\left[\nabla_{x_t} \log p_{\text{robust},\psi}\big(y|f_{\text{dn}}(x_t,t)\big)\right],$$

where $\text{cone}(\alpha,v) := \{w \in \mathbb{R}^d : \angle(v,w) \leq \alpha\}$ is the cone of angle $\alpha$ around vector $v$, and the projection $P_{\text{cone}(\alpha,v)}$ onto $\text{cone}(\alpha,v)$ is given as:

$$P_{\text{cone}(\alpha,v)}[w] := \begin{cases} \langle u,w\rangle u, & \angle(w,v) > \alpha \\ w, & \text{else,} \end{cases}$$

where $P_{v^\perp}(w) := w - \frac{\langle w,v\rangle}{\langle v,v\rangle}v$ and $u = \sin(\alpha)\frac{P_{v^\perp}(w)}{\|P_{v^\perp}(w)\|_2} + \cos(\alpha)\frac{v}{\|v\|_2}$ (note $\|u\|_2 = 1$). The motivation for this projection is to reduce the noise in the gradient of the non-robust classifier. Note

| Original | Non-robust | Robust | Cone Proj. | Original | Non-robust | Robust | Cone Proj. |
|----------|------------|--------|------------|----------|------------|--------|------------|
| ladybug | weevil: 0.99 | weevil: 1.00 | weevil: 0.99 | ringlet | monarch: 0.47 | monarch: 1.00 | monarch: 0.98 |

Figure 3: DVCEs for the non-robust Swin-TF[28], robust MNR-RN50 model, and cone-projected DVCEs for Swin-TF[28]. Similar to adversarial examples, guidance by the non-robust Swin-TF does not yield semantically meaningful changes. In contrast, the DVCEs of the robust MNR-RN50 and the DVCE with cone projection for the Swin-TF[28] yield valid and realistic VCEs.

that the projection of the gradient of the robust classifier onto the cone generated by the non-robust classifier with angle $\alpha < 90°$ is always an ascent direction for $\log p_\phi(y|f_{\mathrm{dn}}(x_t, t))$, which we would like to maximize (note that $g_{\mathrm{proj}}$ is not necessarily an ascent direction for $\log p_{\mathrm{robust},\psi}(y|f_{\mathrm{dn}}(x_t, t))$). Thus, the cone projection is a form of regularization of $\nabla_{x_t} \log p_\phi(y|f_{\mathrm{dn}}(x_t, t))$, which guides the diffusion process to semantically meaningful changes of the image. The averaging over augmentations of $x_t$ has an additional regularizing effect, see Appendix B.6.

### 3.3 Final Scheme for Diffusion Visual Counterfactuals

Our solution for a non-adversarially robust classifier $p_\phi(y|\cdot)$ is to use Algorithm 1 of [14] by replacing the update step with:

$$g_{\mathrm{proj}} = P_{\mathrm{cone}(\alpha, \nabla_{x_t} \log p_\phi(y|f_{\mathrm{dn}}(x_t, t)))}\left[\nabla_{x_t} \log p_{\mathrm{robust},\psi}\left(y|f_{\mathrm{dn}}(x_t, t)\right)\right],$$

$$g_{\mathrm{update}} = C_c \frac{g_{\mathrm{proj}}}{\|g_{\mathrm{proj}}\|_2} - C_d \frac{\nabla_{x_t} d(\hat{x}, f_{\mathrm{dn}}(x_t, t))}{\|\nabla_{x_t} d(\hat{x}, f_{\mathrm{dn}}(x_t, t))\|_2},$$

$$\mu_t = \mu_\theta(x_t, t) + \Sigma_\theta(x_t, t) \|\mu_\theta(x_t, t)\|_2 g_{\mathrm{update}},$$

$$p(x_{t-1}|x_t, \hat{x}, y) = \mathcal{N}(\mu_t, \Sigma_\theta(x_t, t)).$$

For an adversarially robust classifier the cone projection is omitted and one uses $g_{\mathrm{proj}} = \nabla_{x_t} \log p_{\mathrm{robust},\psi}\left(y|f_{\mathrm{dn}}(x_t, t)\right)$. In all our experiments we use $C_c = 0.1$, and $C_d = 0.15$ unless we show ablations for one of the parameters. The angle $\alpha$ for the cone projection is fixed to $30°$. In strong contrast to BD, these parameters generalize across images and classifiers.

## 4 Experiments

In this section, we evaluate the quality of the DVCE. We compare DVCE to existing works in Sec. 4.1. In Sec. 4.2, we compare DVCEs for various state-of-the-art ImageNet models and show how DVCEs can be used to interpret differences between classifiers. For our DVCEs, we use for all experiments the fixed hyperparameters from Sec. 3.3. The diffusion model used for DVCE is from [14] and was trained class-unconditionally on 256x256 ImageNet images using a modified UNet[40]. A user study, discovery of spurious features using VCEs, further experiments, and ablations are in the appendix.

### 4.1 Comparison of Methods for VCE Generation

We compare DVCEs with VCEs produced by BD (BDVCEs) and $l_{1.5}$-SVCEs. As the latter only works for adversarially robust classifiers, we use the multiple-norm robust ResNet50 from [7], MNR-RN50, as the classifier to create VCEs for all three methods. As this model is robust on its own, we do not use the cone projection for its DVCEs.

First, in Fig. 4 we present a qualitative evaluation, where we transform one image into two different classes that are close to the true one in the WordNet hierarchy[33]. The radius of the $l_{1.5}$-ball of [7] is chosen as the smallest $r \in \{50, 75, 100, 150\}$ such that the confidence in the target class is larger than 0.9 per image. For BD, we select the image with the smallest classifier and regularization weight that reaches confidence larger than 0.9 from the set of parameters discussed in Sec. 3.3. If 0.9 is not reached by any setting for one of the two baselines, we show the image that achieves the highest

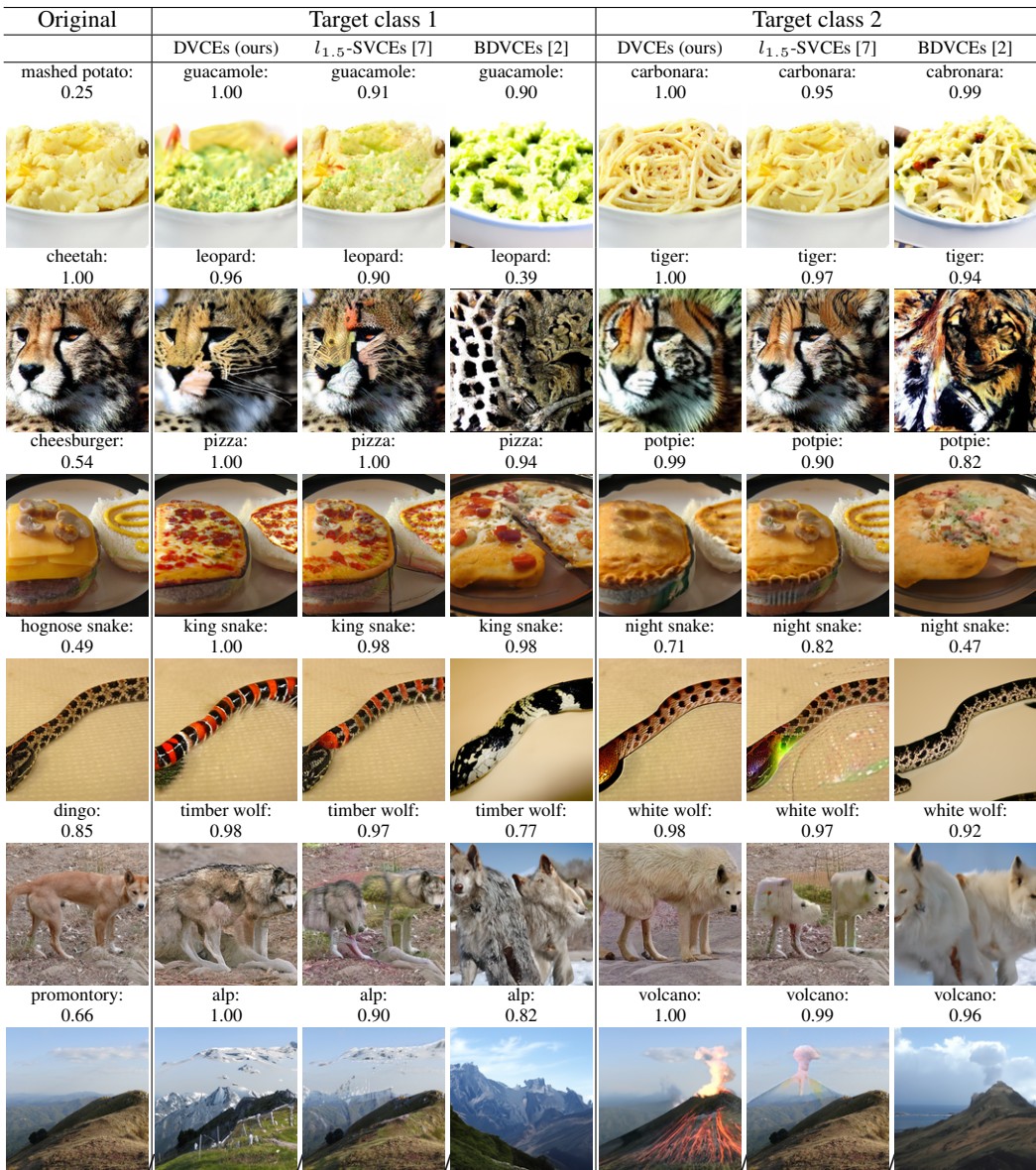

| Original | Target class 1 | | | Target class 2 | | |
|---|---|---|---|---|---|---|
| | DVCEs (ours) | $l_{1.5}$-SVCEs [7] | BDVCEs [2] | DVCEs (ours) | $l_{1.5}$-SVCEs [7] | BDVCEs [2] |
| mashed potato: 0.25 | guacamole: 1.00 | guacamole: 0.91 | guacamole: 0.90 | carbonara: 1.00 | carbonara: 0.95 | carbonara: 0.99 |
| cheetah: 1.00 | leopard: 0.96 | leopard: 0.90 | leopard: 0.39 | tiger: 1.00 | tiger: 0.97 | tiger: 0.94 |
| cheesburger: 0.54 | pizza: 1.00 | pizza: 1.00 | pizza: 0.94 | potpie: 0.99 | potpie: 0.90 | potpie: 0.82 |
| hognose snake: 0.49 | king snake: 1.00 | king snake: 0.98 | king snake: 0.98 | night snake: 0.71 | night snake: 0.82 | night snake: 0.47 |
| dingo: 0.85 | timber wolf: 0.98 | timber wolf: 0.97 | timber wolf: 0.77 | white wolf: 0.98 | white wolf: 0.97 | white wolf: 0.92 |
| promontory: 0.66 | alp: 1.00 | alp: 0.90 | alp: 0.82 | volcano: 1.00 | volcano: 0.99 | volcano: 0.96 |

Figure 4: Comparison of different VCE-methods for the robust MNR-RN50 (taken from [7]). We show for each original image (outer left column) and for each target class our DVCEs (left column), the $l_{1.5}$-SVCEs of [7] (middle), and the adaptation of BDVCEs [2] (right). Only DVCEs satisfy all desired properties of VCEs. BD fails for leopard and tiger and often produces images far away from the original one (pizza, potpie, timber wolf, white wolf, and alp). $l_{1.5}$-SVCEs show artefacts (white wolf, volcano, night snake) and have often lower image quality.

confidence. As Fig. 4 shows, DVCE is the only method that satisfies all desired properties of VCEs. For example, for "mashed potato", DVCE preserves the bowl and only changes the content into either guacamole or carbonara. Our qualitative comparison shows that the same hyperparameter setting of DVCE can handle different classes and transfer between similar classes, such as different snakes or wolf types, as well as different object sizes, e.g. cheetah and snake.

In contrast, both $l_{1.5}$-SVCEs and BDVCEs require different hyperparameters for different images to achieve high confidence in the target class. Even with the six parameter configurations, BD is not able to always produce images with high confidence in the target class. More problematic for VCEs is that the resulting images can have high confidence but are neither realistic (cheetah $\rightarrow$ tiger) nor resemble the original image at all (dingo $\rightarrow$ timber wolf/white wolf). Even if the method works

with the given parameters, for example, mashed potato $\rightarrow$ guacamole or carbonara, the overall image quality cannot match that of DVCE as often the images contain overly bright colors. In the case of the volcano VCE, DVCE shows class features like lava whereas BDVCE can not clearly be labeled as a volcano. For the $l_{1.5}$-SVCEs, one often needs large radii to achieve the desired confidence of 0.9, which often results in images that do not look realistic.

As noted in [7], a quantitative analysis of VCEs using FID scores is difficult as methods not changing the original image have low FID score. Thus, we have developed a cross-over evaluation scheme, where one partitions the classes into two sets and only analyzes cross-over VCEs (more details are in App. E). We show the results in Tab. 1. In terms of closeness, DVCEs are worse than $l_{1.5}$-SVCEs, which can be expected as they optimize inside an $l_{1.5}$-ball. However, DVCE are the most realistic ones (FID score) and have similar validity as $l_{1.5}$-SVCEs. BDVCEs are the worst in all categories.

Moreover, we have conducted a user study (20 users), in which participants decided if the changes of the VCE are **meaningful** or **subtle** and if the generated image is **realistic**, see App. C for details. The percentage of total images having the three different properties is (order: DVCEs, $l_{1.5}$-SVCEs, BD): **meaningful** - **62.0%**, 48.4%, 38.7%; **realism** - 34.7%, 24.6%, **52.2%**; **subtle** - 45.0%, **50.6%**, 31.0%. This confirms that DVCEs generate more meaningful features in the target classes. While the result regarding realism seems to contradict the quantitative evaluation, this is due to fact that realism means that the user considered the image realistic irrespectively if it shows the target class or not.

| Metric | Closeness | | | | Validity | Realism |
|---|---|---|---|---|---|---|
| | $l_1 \downarrow$ | $l_{1.5} \downarrow$ | $l_2 \downarrow$ | LPIPS-Alex $\downarrow$ | Mean Conf. $\uparrow$ | Avg. FID $\downarrow$ |
| DVCEs (ours) | 12799 | 293 | 48 | 0.35 | 0.932 | **17.6** |
| $l_{1.5}$-SVCEs[7] | **5139** | **139** | **26** | **0.20** | **0.945** | 25.6 |
| Blended Diffusion[2] | 35678 | 722 | 108 | 0.58 | 0.825 | 27.9 |

Table 1: Quantitative evaluation of VCEs. DVCEs outperform BDVCEs [2] in all metrics. Moreover, they achieve comparable to $l_{1.5}$-SVCEs Mean Conf. (validity), while outperforming them significantly in Avg. FID. (realism) but do larger changes to the image than SVCE (closeness).

## 4.2 Model comparison

**Non-robust models.** In Fig. 5 we show that DVCEs can be generated for various state-of-the-art ImageNet [41] models. We use a Swin-TF-L[28], a ConvNeXt-L [29] and a Noisy-Student EfficientNet-B7 [57, 53]. Both the Swin-TF and the ConvNeXt are pretrained on ImageNet21k [13] whereas the EfficientNet uses noisy-student self-training on a large unlabeled pool. As all models do not yield perceptually aligned gradients, we use the cone projection with $30°$ angle described in Sec. 3.2 with the robust model from the previous section. All other parameters are identical to the previous experiment. We highlight that DVCE is the first method capable of explaining (in the sense of VCEs) arbitrary classifiers on a task as challenging as ImageNet. Overall, DVCEs satisfy all desired properties of VCEs. It also allows us to inspect the most important features of each model and class. For the stupa and church classes, for example, it seems like all models use different roof and tower structures as the most prominent feature as they spent most of their budget on changing those.

**Robust models.** Here, we evaluate the DVCEs of different robust models trained to be multiple-norm adversarially robust. They are generated by multiple norm-finetuning [11] an initially $l_p$-robust model to become robust with respect to $l_1$-, $l_2$- and $l_\infty$-threat models. Specifically an $l_2$-robust ResNet50 [16] results in MNR-RN50 used in the previous sections, an $l_\infty$-robust XCiT-transformer model [12] - in MNR-XCiT in the following and an $l_\infty$-robust DeiT-transformer [5] - in MNR-DeiT. Their multiple-norm robust accuracies can be found in [11]. In [7], different $l_p$-norm adversarially robust models were compared for the generation of VCEs and they showed that for their $l_{1.5}$-SVCEs the multiple norm robust model was better (both in terms of FID and qualitatively) than individual $l_p$-norm robust classifiers. This is the reason why we also use a multiple-norm robust model for the cone-projection of a non-robust classifier. In Fig. 6, we show DVCEs of the three different classes for two examples images and two target classes each. All of the multiple-norm robust models have similarly good DVCEs showing classifier-specific variations in the semantic changes. This proves again the generative properties of adversarially robust models, in particular of robust transformers. More examples and further experiments are in App. B.4.

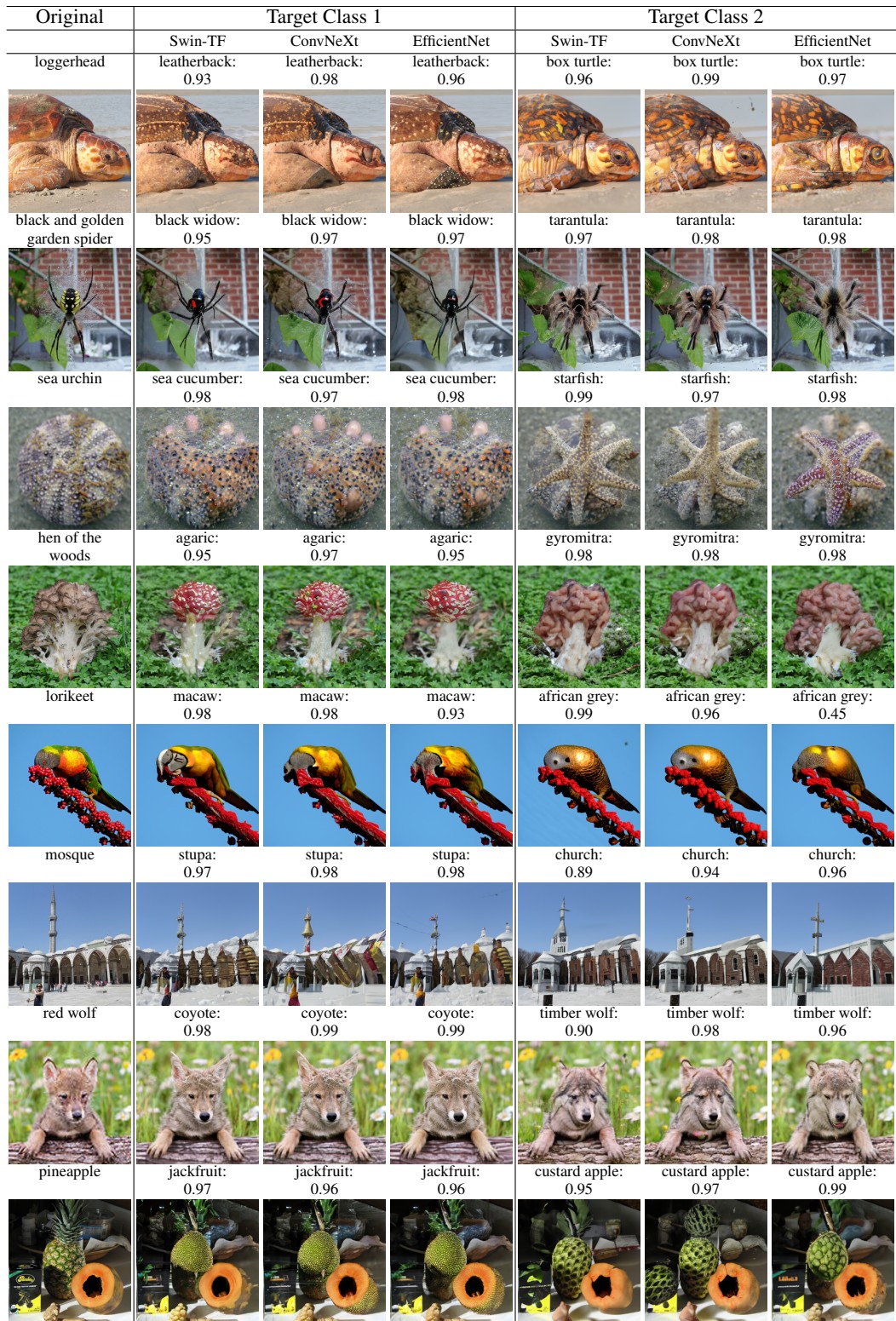

Figure 5: DVCEs for three non-robust classifiers: Swin-TF[28], ConvNeXt[29] and EfficientNet[57]. For each original image (outer left column) we show DVCEs into two target classes. Please zoom into the images to see more fine-grained details and differences.

| Original | Target Class 1 | | | Target Class 2 | | |
|---|---|---|---|---|---|---|
| | MNR-RN50 | MNR-XCiT | MNR-DeiT | MNR-RN50 | MNR-XCiT | MNR-DeiT |
| pirate ship | liner ship: 1.00 | liner ship: 1.00 | liner ship: 1.00 | container ship: 1.00 | container ship: 1.00 | container ship: 1.00 |
| mashed potato | dough: 1.00 | dough: 1.00 | dough: 1.00 | carbonara: 1.00 | carbonara: 1.00 | carbonara: 1.00 |

Figure 6: We compare DVCEs for three different robust models (no cone projection) which are all fine-tuned to be multiple-norm adversarially robust [11], that is against $l_1$, $l_2$ and $l_\infty$-perturbations. More examples can be found in Fig. 10.

## 5 Limitations, Future Work and Societal impact

In comparison to GANs, diffusion-based approaches can be expensive to evaluate as they require multiple iterations of the reverse process to create one sample. This can be an issue when creating a large amount of VCEs and makes deployment in a time-sensitive setting challenging.

We also rely on a robust model during the creation of VCEs for the cone projection. While we show that standard classifiers do not yield the desired gradients, training robust models can be challenging. An interesting direction for future research is thus to replace the cone projection with another "denoising" procedure for the gradient of a non-robust model.

From a theoretical standpoint, using conditional sampling with reverse transitions of the form (7) is justified, in practice, however, we have to approximate the reverse transitions with shifted normal distributions of the form (9). This approximation relies on the conditioning function having low curvature, and it can be hard to verify this once we start adding a classifier, distance and other possible terms and it is unclear how this influences the outcome of the diffusion process.

The quantitative evaluation of VCEs is difficult, as the standard FID metric compares the distribution of features of a classifier over a test and a generated dataset. However, for VCEs it is not only important to generate realistic images, but also to achieve high confidence and to create meaningful changes. Moreover, metrics such as IM1, IM2 [30] for VCEs rely on a well-trained (V)AE for every class, which is difficult to achieve for a dataset with 1000 classes and high-resolution images. Future research should therefore try to develop metrics for the quantitative evaluation of VCEs and we think that the evaluation in Tab. 1 is the first step in that direction.

DVCEs and VCEs in general help to discover biases of the classifiers and thus have a positive societal impact, however one can abuse them for unintended purposes as any conditional generative model.

## 6 Conclusion

We have proposed DVCEs, a novel way to create VCEs for any state-of-the-art ImageNet classifier using diffusion models and our cone projection. DVCEs can handle vastly different image configurations, object sizes, and classes and satisfy all desired properties of VCEs.

## Acknowledgement

The authors acknowledge support by the DFG Excellence Cluster Machine Learning - New Perspectives for Science, EXC 2064/1, Project number 390727645 and the German Federal Ministry of Education and Research (BMBF): Tübingen AI Center, FKZ: 01IS18039A as well as the DFG grant 389792660 as part of TRR 248.

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
