# OpenReview forum: "Diffusion Visual Counterfactual Explanations"
_NeurIPS.cc/2022/Conference — NeurIPS 2022 Accept_

### Official Review · Reviewer_j44j · 2022-07-12

**Rating:** 5
**Confidence:** 3
**Soundness:** 3 good
**Presentation:** 3 good
**Contribution:** 2 fair

**Summary:**

This paper introduces a method to generate visual counterfactual examples (VCEs) using a diffusion model. Given a classifier we would like to explain, the method generates counterfactuals that are realistic and minimal perturbations of a query image and change the query's predicted class. Realism is obtained using a diffusion model as an image prior. The method is similar to other VCEs except using a diffusion model to enforce realism, and the main contributions are the specific techniques used to guide a diffusion model toward generated VCEs.

**Questions:**

My main questions and suggestions are the following:

1. I did not find the evaluation to be completely convincing. I would like to see stronger quantitative evidence that DVCEs are better explanations than the baselines/ablations. This could be quantitative measures of validity/realism/minimality, more convincing user studies, or other metrics that get more at the heart of the "usefulness" of the explanations.

2. I would be interested in seeing Section 4.2 greatly expanded. I think the big thing missing from the paper is a demonstration that we can use DVCEs to actually get insight into different classifiers -- see their biases, understand their weaknesses, pick which of two is better, etc. Section 4.2 barely scratches the surface of these questions, but it is a good start.


**Limitations:**

The limitations are adequately discussed.

**Strengths And Weaknesses:**

I found this to be an interesting paper. The method is sensible and the results are decent. I found the evaluation partially convincing, but with some weaknesses. I will elaborate on these dimensions below:

**Originality**
The main contributions of this paper are in the details. The paper takes two fairly well-established things, VCEs and diffusion models, and puts them together. The exact way they are combined seems reasonable, but also not all that different than other work that uses classifiers or score functions of various kinds to guide generative models. At a glance, the results look quite similar to text/classifier-guided image manipulation. Nonetheless, the target application is slightly different than general image manipulation (rather it is VCE generation) and the specific details (adaptive parameterization, cone-projection) are, to my knowledge, novel.

**Quality**
The method appears to be sound, and the quality of the results is high, at least along the three criteria the paper targets -- validity, realism, minimality. The qualitative comparisons do seem to show the advantages of DVCEs over baselines and ablations, but I'm wary of reading too much into qualitative results. The user study was not terribly convincing -- DVCEs only win on one out of three metrics.

**Clarity**
There are two issues with the writing that could be improved:
1. The math was hard for me to follow. This is in part because the derivations refer to a lot of results from previous works. I would appreciate if the math could be made more self-contained. It also feels like the method is actually pretty simple and could be stated more plainly, e.g., Eqn 10 could be explained as: "take a step in the direction that increases the classifier score while not moving too far away from $\hat{x}$" (this is just an example, my point is just that some of the mathiness could be translated into plainer English, alongside the formal derivations). Equation 6 was especially hard for me to parse and I'm not sure I understand what is intended (although I think I get how $f_{\text{dn}}$ works).
2. In my opinion, the paper somewhat overclaims the contributions. There are three primary ways. First, it is emphasized a few times (e.g., Line 22) that DVCEs are model-agnostic and can be applied to any classifier, and that this distinguishes them from past work. However, results are only shown on ImageNet classifiers, and, as the paper mentions, a prereq is having a diffusion model and robust classifier for the domain. This was not quite what I expected from a method described as "model-agnostic" and that (line 56) "overcomes" the need for retraining for each new classifier (retraining is still needed if you deviate from ImageNet). Second, the success of the user study is overclaimed. Line 68 says DVCEs outperform baselines on the user study, but in fact out of three user study metrics DVCEs were best on just one. The paper argues that, taken holistically, the user study shows that DVCEs are best, but I wouldn't say that's home run evidence. Third, the paper claims (e.g., line 156) that DVCEs "can be applied to any image classifier regardless of whether it is adversarially robust or not." I find this slightly misleading, since a robust classifier is still required. It could be rephrased as something like "can be applied to any image classifier as long as you are given a robust image classifier for that same task."

**Significance**
I believe the paper studies an important topic, and contributes a tool that indeed seems useful. However, in some ways the paper is a step back from other recent work in this area. For example, whereas DVCEs only show one counterfactual, [21] shows multiple counterfactuals, along different visual dimensions. The DVCE results look very similar to text-guided image editing, and indeed this is apparent in the use of an image editing paper, [2], as one of the baselines. This raises the question: what exactly distinguishes the kinds of VCEs presented here from the kind of results in papers like [2] (which did not target counterfactual explanation). The contribution would be clearer if there were metrics or applications of DVCEs that were specific to the goal of _explaining_ classifiers. What do we learn about different classifiers from these visualizations? Can our gain in understanding be measured? These questions are barely addressed by the present paper. From a practical standpoint, the applicability of the method is limited by the requirement of a robust classifier for the task and a good diffusion model for the domain.

---

> ### Author Response · Authors · 2022-08-02
> **Response**
>
> Thank you for providing feedback and taking the time to review our paper.
>
> ---
>
> *“Whereas DVCEs only show one counterfactual, [21] shows multiple counterfactuals, along different visual dimensions.”:*
>
> While [21] (now ref. [28] in the updated version) shows multiple counterfactuals per image, it has an important limitation - that it has been shown to work mostly on binary classification problems and it doesn’t work on ImageNet which is considerably more complex (as also acknowledged by **Reviewer W3sJ** and **BaB4**).
> Moreover, by changing seed, DVCEs can as well generate diverse counterfactuals, as can be seen in the Appendix B.3. To generate more diverse images, one could decrease the regularization weight $C_d$ or increase $T_{\mathrm{start}}$.
>
> *“The DVCE results look very similar to text-guided image editing, and indeed this is apparent in the use of an image editing paper, [2], as one of the baselines”:*
>
> It is true that one can generate classifier-guided images in general. The challenge lies in generating classifier-guided images that are close to the original one (minimality/closeness) and increase the confidence of the classifier significantly (validity), as non-robust models can be fooled by adversarial perturbations, and generative models, such as diffusion model, can generate natural-looking adversarial examples, that don’t add discernible features of the target class, but increase the confidence significantly, as can be seen in Figure 3. Moreover, as we illustrated for [2] the problem is that for every image one needs to choose different hyperparameters to get good results (if they exist at all). This is clearly not practical and our approach overcomes this as well.
>
> *“From a practical standpoint, the applicability of the method is limited by the requirement of a robust classifier for the task and a good diffusion model for the domain.”:*
>
> Yes, this is true, but for most image classification tasks, we now have robust classifiers (or it is straightforward to create one using adversarial training) and diffusion models are becoming increasingly popular so that they will be soon available for much more datasets. Thus we don’t see this as an obstacle.
>
> *“Some of the mathiness could be translated into plainer English, alongside the formal derivations.”:*
>
> Thank you for your suggestion! We have added an intuition for Eq. 10 in Lines 139-140. We have moreover made Eq. 6 more convenient to parse. Furthermore, Eq. 6 describes the mapping for the denoising step of every sample before feeding it into the classifier, following [2].
>
> ---
>
> **Questions:**
>
> Q1: *“I would like to see stronger quantitative evidence that DVCEs are better explanations than the baselines/ablations.”*
>
> We agree. Please see our general comment regarding quantitative evaluation.
>
> Q2: *“What do we learn about different classifiers from these visualizations? Can our gain in understanding be measured?” ,  “I would be interested in seeing Section 4.2 greatly expanded. I think the big thing missing from the paper is a demonstration that we can use DVCEs to actually get insight into different classifiers -- see their biases, understand their weaknesses, pick which of two is better, etc. Section 4.2 barely scratches the surface of these questions, but it is a good start.”*
>
> We agree that this is ultimately the goal and admittingly we have focused in the paper on the fact that we can generate minimal, realistic, and valid VCEs for different state-of-the-art ImageNet classifiers for full ImageNet.
> We think that Section 4.2 already shows some differences but we think that a full-blown analysis would require its own paper. However, we see your point and thus have instead investigated in the novel Appendix F the search for potential spurious features of the classifiers. We recover the spurious features found in the previous work using SVCEs and also find novel ones using our DVCEs.

---

> > ### Comment · Reviewer_j44j · 2022-08-09
> > **thanks for the responses**
> >
> > Thanks for the detailed responses! The new quantitative experiments are good to see, and I appreciate the effort in Appendix F. I will consider raising my score; I think the paper is improved and don't have additional questions at this time.

---

### Official Review · Reviewer_djLx · 2022-07-12

**Rating:** 6
**Confidence:** 4
**Soundness:** 4 excellent
**Presentation:** 3 good
**Contribution:** 3 good

**Summary:**

The authors propose a framework to generate visual counterfactual explanations using diffusion processes. In order to generate non-trivial and close to input CFs, the authors modify the diffusion process accordingly: by introducing a distance function in the diffusion process and the choice of the starting point, adaptive re-parameterization, and cone projection of the gradient (using a robust classifier) as regularization.

**Questions:**

- I suggest the authors discuss the fail cases of the method and show some examples.
- I suggest the authors add a closely related work [1] in their discussions
- I suggest the authors add Supp B.4 to the main paper.

[1] Khorram and Fuxin. Cycle-Consistent Counterfactuals by Latent Transformations. CVPR 2022.


**Limitations:**

The authors have discussed the limitations.

**Strengths And Weaknesses:**

Strengths:
- The authors focus on an important topic of generating realistic CF examples.
- The paper is well-written and easy to follow.
- The authors have made novel contributions by adapting diffusion models into generating CF images.
- The experiments show the effectiveness of the proposed method against baselines.

Weaknesses:
- A key factor in generating realistic CFs using the proposed method is the existence of an adversarially robust model which is not often the case.
- No quantitative comparison in terms of the realism/validity/closeness of the generated CFs.

---

> ### Author Response · Authors · 2022-08-02
> **Response**
>
> Thanks a lot for the detailed review and the helpful suggestions to improve the paper. We are happy that the reviewer thinks that we “focus on the important topic of generating realistic CF examples” and that we “made novel contributions by adapting diffusion models into generating CF images”. We address the weaknesses/questions in the following:
>
> ---
>
> *“A key factor in generating realistic CFs using the proposed method is the existence of an adversarially robust model which is not often the case.”:*
>
> For most image classification problems adversarially robust models are available. If not, then using adversarial training they are easy to train from scratch or one can fine-tune a robust Imagenet model. Please, see also the discussion with the **Reviewer j44j**.
>
> *“No quantitative comparison in terms of the realism/validity/closeness of the generated CFs.”:*
>
> Please see our general comment.
>
> ---
>
> **Questions:**
>
> Q1: *“I suggest the authors discuss the fail cases of the method and show some examples. ”*
>
> Please see the novel Appendix G for the discussion of failure cases: i) sometimes it happens that the VCEs are slightly blurry which seems an artefact of the diffusion model as it happens for blended diffusion and is visible also in the original diffusion paper (see top left image in Figure 15 of [15]), ii) our DVCE can fail when the change is difficult to realize (i.e. the original image is not similar to the target class).
>
> Q2: *“I suggest the authors add a closely related work [A] in their discussions.”*
>
> Thanks a lot for the reference. We discuss it now in the introduction. A direct comparison would be desirable but code is only available for the experiments on MNIST/FMNIST. The code to reproduce the ImageNet experiments is not available and it was unclear to us if the method uses a classifier for the generated images for the ImageNet classes (they show only five examples) or if only the conditional generative model is used.
>
> Q3: *“I suggest the authors add Supp B.4 to the main paper.”*
>
> Thank you for the excellent suggestion! We will add several examples from the Appendix B.4 to the main paper in the final version.
>
> **References used for this part of the rebuttal:**
>
> [A] Khorram and Fuxin. Cycle-Consistent Counterfactuals by Latent Transformations. CVPR 2022.

---

### Official Review · Reviewer_BaB4 · 2022-07-14

**Rating:** 7
**Confidence:** 3
**Soundness:** 3 good
**Presentation:** 3 good
**Contribution:** 3 good

**Summary:**

The paper focuses on model-agnostic visual counterfactual explanation (VCE). The proposed method builds on the diffusion model and generates counterfactual samples for the target class with minimal changes. The authors first handle the robust classifier by using adaptive parameterization. And then they solve the non-robust classifier by projecting the gradient of a robust classifier onto the cone of the non-robust classifier. They experiment on ImageNet classifier and demonstrate qualitatively for dozens of examples.

**Questions:**

1. How do you select the project angle (30 degrees)? Is it empirically determined?
2. Is $\Sigma_{\theta}$ diagonal? If so, does that mean each feature/pixel of the generated sample is independent?
3. With the projection of the gradient for the non-robust classifier, how hard is the training?

**Limitations:**

See the above.

**Strengths And Weaknesses:**

Strengths:
Although it is not new to use the diffusion model for generating counterfactual samples, the proposed method is new in the sense that it has no constraint on the classifier to be explained. In addition, the proposed method can handle a bigger class set. The qualitative results support their claim overall.

Weaknesses:
It seems to depend on a robust classifier anyway. As stated in the paper, the computational expense might be high because of the time-related generative process. More importantly, there are no quantitative comparisons between models, which makes it hard to evaluate. Although it is hard to design quantitative metrics for VCE, it is still possible to measure validity and closeness.

---

> ### Author Response · Authors · 2022-08-02
> **Response**
>
> Thanks a lot for your review and your questions. We are happy that the reviewer acknowledges that we can generate semantically meaningful visual counterfactual explanations (VCEs) for general classifiers and are not restricted to a subset of the ImageNet classes.
>
> ---
>
> *“It seems to depend on a robust classifier anyway.”*
>
> Our cone projection allows us to explain any classifier and produce VCEs that achieve high confidence in the target class for the target model but also visually change the image in a plausible fashion. As we show in Figure 3, even in combination with a diffusion model, the adversarial vulnerabilities of non-robust models prevent proper guidance of the diffusion process. Thus we need a robust classifier to guide the process
>
> *“As stated in the paper, the computational expense might be high because of the time-related generative process.”*
>
> Thanks for asking this question. In the paper, we forgot to mention the number of diffusion steps.
> The number of total diffusion steps in our work is set to $T=200$ (added in line 157), however as we start our diffusion process at $T/2$ (Section 3.1), we only require 100 effective diffusion steps. To achieve classifier guidance, the forward-backward pass through the target model is required plus the gradient computation for the robust model. The cone projection itself is negligible from a computational standpoint. Overall, the computational complexity of our method is similar to that of previous works using diffusion.
>
> “More importantly, there are no quantitative comparisons between models, which makes it hard to evaluate. Although it is hard to design quantitative metrics for VCE, it is still possible to measure validity and closeness. “
>
> Please see our general comment regarding quantitative evaluation.
>
> ---
>
> **Questions:**
>
> Q1: *“How do you select the project angle (30 degrees)? Is it empirically determined?”*
>
> Yes, an angle of 30 degrees is typically sufficient to get semantically meaningful changes. We introduced an additional Ablation in Appendix B.5 which illustrates the effect of different angles on the outcome. In short, too small angles do not produce visually-meaningful changes and often fail to achieve high confidence for the target model in the desired class. Very large angles work, however at that point we allow the method to strongly deviate from the target model’s gradient, which is undesirable as we want to explain the target and not the robust model.
>
> Q2: *“Is $\Sigma_\theta$ diagonal? If so, does that mean each feature/pixel of the generated sample is independent?”*
>
> The diffusion model we use is from “Diffusion Models Beat GANs on Image Synthesis” and uses diagonal covariances. Note that this is theoretically justified, as [48] have shown that for a Gaussian diffusion process, the reverse transitions $q(x_{t-1} | x_t)$ approach *diagonal* Gaussian distributions as the number of total diffusion steps $T$ approaches infinity.
> However, this does not yield independent pixels in the final image. At every timestep $t$ given $x_{t}$, the next sample $x_{t-1}$ is sampled from $\mathcal{N}(\mu_\theta(x_t,t), \Sigma_\theta(x_{t},t))$ (this argument works for diffusion processes with or without a guidance).  But both $\mu$ and and $\Sigma$ are parameterized using an image-to-image DNN which connects each input to each output pixel. This means that every feature in $x_{t-1}$ is dependent on every pixel in $x_{t}$, therefore the pixels in the generated sample $x_{0}$ are not independent.
>
> Q3: *“With the projection of the gradient for the non-robust classifier, how hard is the training?”*
>
> We might misunderstand the question, but there is no additional training necessary. We use the classifier as it is. The robust model is trained with standard adversarial training and the target classifier can be trained in any way and does not require additional training/fine-tuning. The diffusion model also remains unchanged from the original formulation. Thus, the cone-projection only appears in the final DVCE sampling and not during model training.

---

### Official Review · Reviewer_W3sJ · 2022-07-16

**Rating:** 7
**Confidence:** 3
**Soundness:** 3 good
**Presentation:** 3 good
**Contribution:** 3 good

**Summary:**

This paper focus on generating Visual Counterfactual Explanations (VCE) for any arbitrary classifier using diffusion models. Prior work generates VCE reliably on adversarial robust models or generative models limited to smaller number of classes. This work leverages diffusion models to generate realistic and minimal semantically altered VCE examples. The authors propose to condition the generated image on the target class using the desired classifier and condition on the original image with L1 distance regularization. The parameterization of such formulation allowed to fix the hyperparameters across different images and models i.e. no need to finetune the hyperparameters of diffusion process for different images and models. Inspired by prior works that shown VCEs are reliably generated on adversarially robust models, authors leverage such robust models and align the gradients of non-robust models with robust models using cone projection. Such projection enabled to generate VCEs reliably for arbitrary classifier. Qualitative results show that the proposed method generates VCE better than the existing works.

**Questions:**

Please refer Weakness above.

**Limitations:**

Authors addressed the limitations and negative impact.

**Strengths And Weaknesses:**

Strengths:
1)	Paper is well written and motivation of the problem formulation is clearly explained.
2)	Formulation for the class conditioning and distance regularization is clear.
3)	Proposed cone projection of the gradients of non-robust classifier on to the gradients of robust classifier helped to generate semantically meaningful VCEs for non-robust classifiers.
4)	Generated VCEs are semantically meaningful and perceptually looks improved than the baselines.

Weakness:
1)	Generally lacking a quantitative measure to evaluate the generated VCEs. Evaluation is mainly performed with visual inspection.
2)	As the integration of cone projection shown to be helpful, however it is not clear why this particular projection is chosen. Are there other projections that are also helpful? Is there a theoretical proof that this cone projection resolves the noise of the gradients in non-robust classifiers?

Overall, I think the proposed technique yield better VCE and interesting for the community. I also think that the strengths outweigh the weakness. However, I would be open to hear other reviewers opinion here.

---

> ### Author Response · Authors · 2022-08-02
> **Response**
>
> Thanks a lot for the detailed review and the interesting questions. We are grateful that the reviewer acknowledges that we can generate semantically meaningful visual counterfactual explanations (VCEs) for general non-robust models and for the full set of ImageNet classes. We address in the following the mentioned weaknesses/questions.
>
> ---
>
> W1: *Lack of Quantitative measures*
>
> Please see our general comment.
>
> W2a): *“As the integration of cone projection is shown to be helpful, however, it is not clear why this particular projection is chosen. Are there other projections that are also helpful?”*
>
> In Figure 3, we show that state-of-the-art ImageNet models are not able to produce VCEs that visually transfer the image into the target class due to their adversarial vulnerability. Adversarially robust models are known to have perceptually aligned gradients [1, 7], however, in our work, we want to be able to also produce VCEs for general non-robust models. The cone projection allows the robust model to guide the non-robust model to produce visually meaningful changes while keeping the overall direction of the target model’s gradient to achieve high confidence. In particular, the cone-projection is always guaranteed to be an ascent direction with respect to the confidence of the target model (would not be the case if we would project the gradient of the target model on the cone generated by the gradient of the adversarially robust model) which was the main guiding principle for us.
>
> W2b) *“Is there a theoretical proof that this cone projection resolves the noise of the gradients in non-robust classifiers?”*
>
> We would argue that there is only limited theoretical understanding (in the sense of proof) of why adversarially robust models have more meaningful gradients than standard models. Thus we doubt that it is easy to come up with proof that the cone projection removes the noise. However, we agree that this is a very interesting question for future research but which is out of scope for this paper.

---

> > ### Comment · Reviewer_W3sJ · 2022-08-10
> > **Thanks for the response**
> >
> > Authors have partially addressed my concerns and I am willing to maintain my score.

---

### Author Response · Authors · 2022-08-02
**General comment**

# Quantitative Evaluation
All reviewers raised the question about quantitative evaluation. As also acknowledged by some reviewers, a quantitative evaluation of visual VCEs is difficult (note that we not only have to generate semantically meaningful images (realistic) but they should also be close to the original image (closeness) and have high confidence (validity)). Thus a single number will not describe this multi-objective problem well. During the preparation of the paper, we tried the evaluation using the IM1/IM2 metrics of [A] which works for MNIST/FMNIST but did not give meaningful results for ImageNet and thus did a user study instead.

However, we clearly agree that a quantitative evaluation strengthens our qualitative evaluation. Thus we evaluate closeness in terms of $l_1, l_{1.5}, l_2$ metrics as well as LPIPS as a more perceptual metric for the robust Madry model (note that SVCEs cannot be applied to non-robust models). Moreover, we report the mean confidence as well as FID (Fréchet Inception Distance) scores for VCEs generated by the following cross-over experiment: i) we split the ImageNet classes into two disjoint groups A and B (by splitting each WordNet category of similar classes), ii) generate VCEs for the ImageNet validation set for A resp. B with target classes from B resp. A (targets are in the same WordNet category) and then compute the FID scores for the VCEs with targets in A resp. B with respect to the training data of the group of the classes of A resp. B. In this way, we overcome the problem that FID scores can be easily optimized by not changing the image at all.


| Metric                                      | $l_{1}$         | $l_{1.5}$      | $l_{2}$       | LPIPS-Alex     | Mean Conf.     | Avg. FID    |
|---------------------------------------------|-----------------|----------------|---------------|----------------|----------------|-------------|
| DVCEs (ours)                                | 12798.8         | 293.4          | 48.4          | 0.352          | 0.932          | **17.6** |
| Blended diffusion | 35677.9         | 722.4          | 107.9         | 0.579          | 0.825          | 27.9        |
| $l_{1.5}$-SVCEs     | **5139.0** | **139.3** | **25.5** | **0.204** | **0.945** | 25.6        |


In terms of distance measures ($l_1, l_{1.5}, l_2$, and LPIPS) it is not surprising that SVCE perform best as they directly maximize confidence in the target class for a given $l_{1.5}$-ball (here radius 150). Our DVCEs have almost the same mean confidence (same validity)  as SVCE but are further away in distance with respect to all metrics. However, our DVCEs have a much lower FID score than SVCE which confirms the qualitative picture that the changes of DVCE are more meaningful and show fewer artefacts (realism). Blended diffusion is the worst in all categories (even though we give it the advantage compared to DVCE/SVCE that they can generate two VCE and we take the one with higher confidence). This is again due to greatly varying results depending on the chosen hyperparameters which shows the advantage of our method.

The low FID score of blended diffusion seemingly contradicts the user study where blended diffusion has been considered best in realism but  one has to note that the users were asked (“Which images look realistic?”). The problem is that blended diffusion does sometimes not change the images significantly (in which case they still look realistic) or they are changed drastically and to a good-looking image but not necessarily to the correct class - both of these hurt the FID score in our “crossover” VCE generation setting. However, note that DVCEs in the user study were best in the “meaningful change” which also explains its better FID score.

In the final version, we will move these quantitative results to the main part. The modifications of the paper compared to the original paper are marked in blue (most of it is in the Appendix).

**References used for this part of the rebuttal:**

[A] Arnaud Looveren and Janis Klaise, Interpretable counterfactual explanations guided by prototypes, 2019.

---

### Meta-Review · Area_Chair_2zCt · 2022-08-28

**Recommendation:** Accept
**Confidence:** Less certain

**Metareview:**

On the whole after the reviews and discussion, the reviewers are in agreement that this is an interesting method with some novel contributions and high-quality results, which should be of interest to the community.

**Award:**

No

---

### Decision · Program_Chairs · 2022-09-14

Accept